# Factors associated with lenvatinib adherence in thyroid cancer and hepatocellular carcinoma

Yoshikazu Tateai[1]*, Kazuyoshi Kawakami[1], Minori Teramae[2], Naoki Fukuda[3], Takashi Yokokawa[1], Kazuo Kobayashi[1], Naoki Shibata[1], Wataru Suzuki[1], Hisanori Shimizu[1], Shunji Takahashi[3], Masato Ozaka[4], Naoki Sasahira[4], Satoko Hori[2], Masakazu Yamaguchi[1]

1 Department of Pharmacy, Cancer Institute Hospital, Japanese Foundation for Cancer Research, Tokyo, Japan, 2 Division of Drug Informatics, Keio University Faculty of Pharmacy, Tokyo, Japan, 3 Department of Medical Oncology, Cancer Institute Hospital, Japanese Foundation for Cancer Research, Tokyo, Japan, 4 Department of Hepato-Biliary-Pancreatic Medicine, Cancer Institute Hospital, Japanese Foundation for Cancer Research, Tokyo, Japan

* yoshikazu.tateai@jfcr.or.jp

**Data Availability Statement:** The data underlying the results presented in the study are available from (available at Dryad https://datadryad.org/stash/dataset/doi:10.5061/dryad.t1g1jwt86).

## Abstract

### Background

Lenvatinib is an oral anticancer medication used to treat radioiodine-refractory thyroid cancer and unresectable hepatocellular carcinoma. The purpose of this study is to evaluate lenvatinib adherence by patients and to identify factors associated with decreased lenvatinib adherence.

### Methods

Among 153 patients who started treatment with lenvatinib for unresectable thyroid cancer or unresectable hepatocellular carcinoma between May 1, 2015 and August 31 2021 at the Cancer Institute Hospital of the Japanese Foundation for Cancer Research, 102 were eligible for this study (55 thyroid cancer, 47 hepatocellular carcinoma). The lenvatinib adherence rate in a treatment cycle was defined as the number of times a patient took lenvatinib in a 28-day cycle divided by the prescribed 28 doses. The rate was determined by pill counting and self-reporting at the pharmaceutical outpatient clinic. Reasons for non-adherence were established by interview and analyzed.

### Results

The median adherence rate of lenvatinib in the first cycle was 90.1% (n = 55) in thyroid cancer and 94.9% (n = 47) in hepatocellular carcinoma. In thyroid cancer, there were 255 incidents of lenvatinib non-adherence. Non-adherence was mainly associated with bleeding events (18.6%), followed by hand-foot skin reactions (10.6%). In hepatocellular carcinoma, there were 97 incidents of non-adherence. Hypertension accounted for 20.6%, followed by hoarseness (18.6%) and diarrhea (17.5%).

**Funding:** The author(s) received no specific funding for this work.

**Competing interests:** The authors have declared that no competing interests exist.

## Conclusion

The adherence rate for lenvatinib in Japanese patients with thyroid and hepatocellular carcinoma in real-world clinical practice was more than 90% in this study. Hypertension was a major reason for non-adherence, followed by hand-foot skin reactions and diarrhea.

## Introduction

Oral anticancer medications are increasingly common [1], and provide patients with the convenience of home administration and reduced travel and appointment frequency [2]. However, accurately measuring and monitoring medication adherence in cancer patients is critical both in clinical practice and research settings, and continues to be a challenging task globally [3]. It has already been reported that drug adherence is important in patients with atherosclerotic cardiovascular disease [4]. Non-adherence is associated with poor treatment efficacy, greater health care costs, and increased hospitalization among patients with cancer [5].

Lenvatinib is an oral multikinase inhibitor of vascular endothelial growth factor receptors 1–3, fibroblast growth factor receptors 1–4, platelet-derived growth factor receptor-α, ret protooncogene, and stem cell factor receptor [6, 7]. In clinical practice, lenvatinib has been used in patients with radioiodine-refractory thyroid cancer [8], unresectable hepatocellular carcinoma [9], advanced endometrial cancer [10], advanced renal cell carcinomas [11], and advanced or metastatic thymic carcinoma [12]. In the study of lenvatinib in differentiated cancer of the thyroid (SELECT), a phase 3 study of 392 patients with radioiodine-refractory differentiated thyroid cancer, treatment with lenvatinib significantly prolonged progression-free survival compared with placebo [8]. In the phase 3, non-inferiority trial of lenvatinib in unresectable hepatocellular carcinoma (REFLECT), 954 patients were randomly assigned to lenvatinib (n = 478) or sorafenib (n = 476). Median survival time with lenvatinib was non-inferior to that with sorafenib [9]. Therefore, lenvatinib is a key drug for treating thyroid cancer and hepatocellular carcinoma. On the other hand, side effects of lenvatinib include hypertension, fatigue, diarrhea, hand-foot skin reactions, and proteinuria. Therefore, adverse effect management and adherence to lenvatinib are important clinical considerations.

Health insurance in Japan covered lenvatinib for thyroid cancer, hepatocellular carcinoma, thymic carcinoma, endometrial cancer, and renal cell carcinoma, in that order. Patients with thyroid cancer and hepatocellular carcinoma were the subjects in the present study because it was conducted at a time when lenvatinib was commonly used for thyroid cancer and hepatocellular carcinoma. In addition, since the standard doses for thyroid cancer (24 mg/day) and hepatocellular carcinoma (12 mg for 60 kg or more, 8 mg for 60 kg or less) differed, the differences between these two cancer types were investigated.

The aim of this study is to evaluate adherence to lenvatinib in thyroid and hepatocellular carcinoma patients in real-world conditions of oncology practice, as well as to identify factors that may reduce adherence to lenvatinib and to obtain information that would be helpful to increase adherence in the future.

## Methods

### Study setting and population

Patients who started treatment with lenvatinib for unresectable thyroid or unresectable hepatocellular carcinoma between May 2015 and August 31 2021 at the Departments of Medical Oncology and Hepato-Biliary-Pancreatic Medicine, the Cancer Institute Hospital of the

Japanese Foundation for Cancer Research, were eligible for inclusion. Lenvatinib was given at 24 mg/day for thyroid cancer. In the case of hepatocellular carcinoma the dose was 12 mg/day for patients with a body weight of 60 kg or more and 8 mg/day for those with a body weight of 60 kg or less. Depending on the patient's condition, the dose was reduced from the starting dose as appropriate. Loperamide was prescribed as a prophylactic to prevent diarrhea, and moisturizers were prescribed to prevent hand-foot skin reactions. Prophylactic drugs for hypertension were not administered. The criteria for dose reduction and withdrawal were as follows. In cases of Grade 2 hypertension, antihypertensive drugs were to be administered; in cases of Grade 3 hypertension despite antihypertensive treatment, the drug was to be stopped and then re-started at a reduced dose. In cases of Grade 4 hypertension, the drug was to be withdrawn. In cases of hematological toxicity and Grade 3 proteinuria, the drug was to be withdrawn until the patient recovered to the state before the start of treatment or Grade 2 or less. When resuming administration, the dose was not to be reduced at the first occurrence of an adverse reaction, but the dose was to be reduced at the second and subsequent occurrences of adverse reactions. If other Grade 2 or Grade 3 adverse reactions that were not tolerated were observed, the drug was to be withdrawn until the patient recovered to the state before starting or to Grade 1 or below. If Grade 4 adverse reactions occurred (in the case of non-life-threatening laboratory test abnormalities, the same treatment as for Grade 3 adverse reactions), drug administration was to be discontinued.

The lenvatinib adherence rate in a cycle was defined as the number of times a patient took lenvatinib in one 28-day cycle divided by the prescribed 28 doses. The first dose taken on day 1, then days 1–28 comprised the 1st cycle, days 29–56 the 2nd cycle, and days 57–84 the 3rd cycle. Relative dose intensities (RDI) [13] and other data were obtained retrospectively from electronic records. The RDI was calculated from the prescription history and pharmacist records as (standard dose—physician prescribed dose reduction—patient self-withdrawal dose) divided by the standard dose.

## Data collection

We collected age, gender, cancer type, Eastern Cooperative Oncology Group (ECOG) performance status (PS), starting dose, residence, treatment line, number of concomitant medications, reason for non-adherence to lenvatinib, and reason for changing or discontinuing treatment. Adherence to lenvatinib was assessed using pill counting and self-reporting treatment journals, and non-adherent patients were interviewed to establish their reasons for not taking lenvatinib at every visit to the pharmaceutical outpatient clinic. If there were multiple reasons, we asked which had had the greatest impact. To determine the pill count, any unused pills were counted during visits to a pharmaceutical outpatient clinic. All participants in this study visited a pharmaceutical outpatient clinic and completed the assessments.

## Pharmaceutical outpatient clinic

In the Pharmaceutical Outpatient Clinic of the Cancer Institute Hospital of the Japanese Foundation for Cancer Research, pharmacists checked adherence to oral anticancer drugs and assessed side effects, which could be a reason for non-adherence, before the patients were examined by their physicians [14, 15]. In particular, the pharmacists played important roles in confirmation and provision of advice. Confirmation involved checking the patients' adherence to lenvatinib and evaluating the side effects. Advice involved including in patients' electronic medical records suggestions regarding the most effective prescription for supportive pharmacotherapy, the timing of the next anticancer drug dose, and comments on the administration regimen.

## Results

### Patients and characteristics

Among 153 patients, patients with adherence follow-up less than 28 days (n = 49), no visit to a pharmaceutical outpatient clinic (n = 22), interruption of visits to a pharmaceutical outpatient clinic (n = 16), suspension of medical treatment in the 1st cycle (n = 9), change of hospital at the 1st cycle (n = 2), or no data collected (n = 2) were excluded, as summarized in Fig 1. Finally, a total of 102 patients were included in the study (Table 1), 55 with thyroid cancer and 47 with hepatocellular carcinoma. The ECOG PS of patients was 0 in 66 patients (64.7%), PS 1 in 21 patients (20.6%), and unknown in 15 patients (14.7%). The starting dose of lenvatinib for thyroid cancer was 24 mg in 36 patients (65.5%), 20 mg in 14 patients (25.4%), and 14 mg in 5 patients (9.1%). The starting dose for hepatocellular carcinoma was 12 mg in 23 patients (49%), 8 mg in 23 patients (49%), and 4 mg in 1 patient (2%). In thyroid cancer, 47 (85.5%) received first-line treatment, 6 (10.9%) second-line treatment, and 2 (3.6%) third-line treatment. In hepatocellular carcinoma, 35 (74.5%) received first-line treatment, 4 (8.5%) second-line treatment, 5 (10.6%) third-line treatment, and 3 (6.4%) fourth-line treatment.

### Adherence rate and RDI

Fig 2 shows the lenvatinib adherence rate in patients with thyroid cancer and hepatocellular carcinoma as determined from the pill counting and self-reporting in the pharmaceutical outpatient clinic. In thyroid cancer, the median adherence rate was 90.1% (n = 55) in the first cycle, 91.4% (n = 42) in the second cycle, and 96.1% (n = 34) in the third cycle. In hepatocellular carcinoma, the median adherence rate was 94.9% (n = 47) in the first cycle, 97.5% (n = 31) in the second cycle, and 97.1% (n = 26) in the third cycle. For thyroid cancer, the median

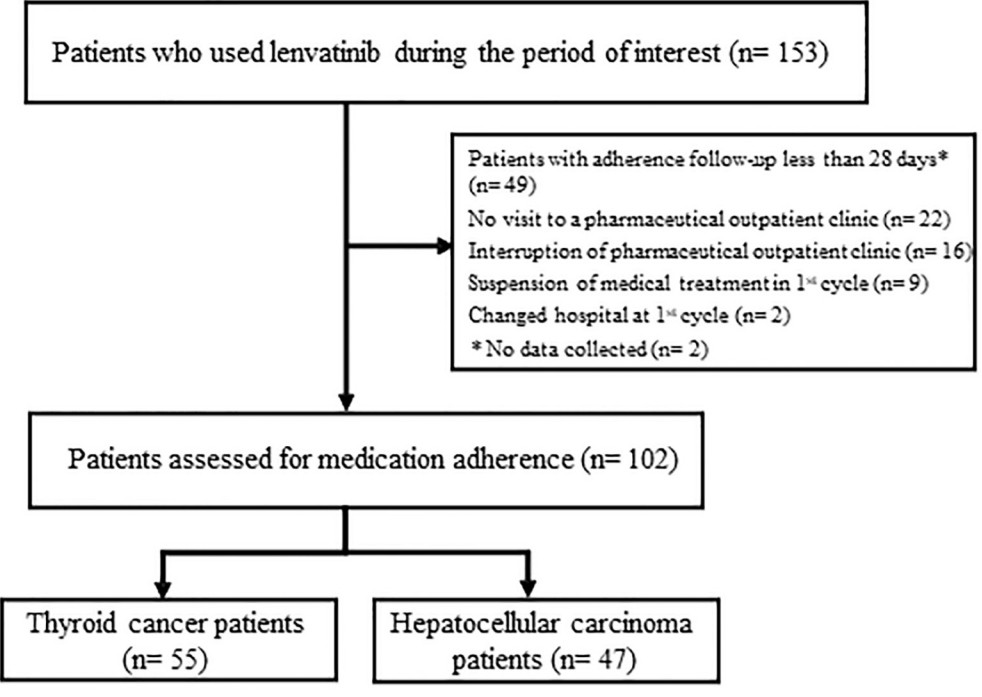

**Fig 1. Flow chart of study patient selection and reasons for exclusion.**

**Table 1. Characteristics of the study population.**

| Characteristic | Number of patients (%) |
|---|---|
| Median age, y (range) | 69 (27–83) |
| Sex | |
| Male / Female | 64 (62.7) / 38 (37.3) |
| Cancer type histology | |
| Thyroid cancer | 55 (53.9) |
| Papillary carcinoma | 37 (36.3) |
| Undifferentiated carcinoma | 9 (8.8) |
| Follicular carcinoma | 5 (4.9) |
| Poorly differentiated carcinoma | 3 (2.9) |
| Medullary carcinoma | 1 (1.0) |
| Hepatocellular carcinoma | 47 (46.1) |
| ECOG performance status | |
| 0 / 1 / unknown | 66 (64.7) / 21 (20.6) / 15 (14.7) |
| Starting dose | |
| Thyroid cancer | |
| 24 mg / 20 mg / 14 mg | 36 (65.5) / 14 (25.4) / 5 (9.1) |
| Hepatocellular carcinoma | |
| 12 mg / 8 mg / 4 mg | 23 (49.0) / 23 (49.0) / 1 (2.0) |
| Residence | |
| Living with someone / living alone / unknown | 81 (79.4) / 20 (19.6) / 1 (1.0) |
| Treatment Line | |
| Thyroid cancer | |
| 1 / 2 / 3 / 4 | 47 (85.5) / 6 (10.9) / 2 (3.6) / 0 (0.0) |
| Hepatocellular carcinoma | |
| 1 / 2 / 3 / 4 | 35 (74.5) / 4 (8.5) / 5 (10.6) / 3 (6.4) |
| Number of concomitant medications | |
| ≥6 / <6 | 20 (19.6) / 82 (80.4) |

Total number of patients N = 102, ECOG: Eastern Cooperative Oncology Group

adherence rate was 90.1% (n = 55) in cycle 1, 91.4% (n = 42) in cycle 2, and 96.1% (n = 34) in cycle 3. In hepatocellular carcinoma, the median adherence rate was 94.9% (n = 47) in cycle 1, 97.5% (n = 31) in cycle 2, and 97.1% (n = 26) in cycle 3. The median relative dose intensity of lenvatinib was 66.3% in thyroid cancer and 76.2% in hepatocellular carcinoma.

## Factors reducing lenvatinib adherence

Fig 3 summarizes the factors associated with lenvatinib non-adherence in patients with thyroid cancer and hepatocellular carcinoma. In thyroid cancer, there were 255 incidents of lenvatinib non-adherence. Bleeding-related events accounted for 18.6% (46 events), followed by hand-foot skin reaction in 10.6% (27 events) and abdominal pain in 8.6% (22 events). In hepatocellular carcinoma, there were 97 incidents of lenvatinib non-adherence. Hypertension was the most common factor, accounting for 20.6% (20 events), followed by hoarseness (18.6%; 18 events) and diarrhea (17.5%; 17 events). Common factors associated with non-adherence in both cancer types were fatigue [7.8% (20 events) for thyroid cancer and 6.2% (6 events) for

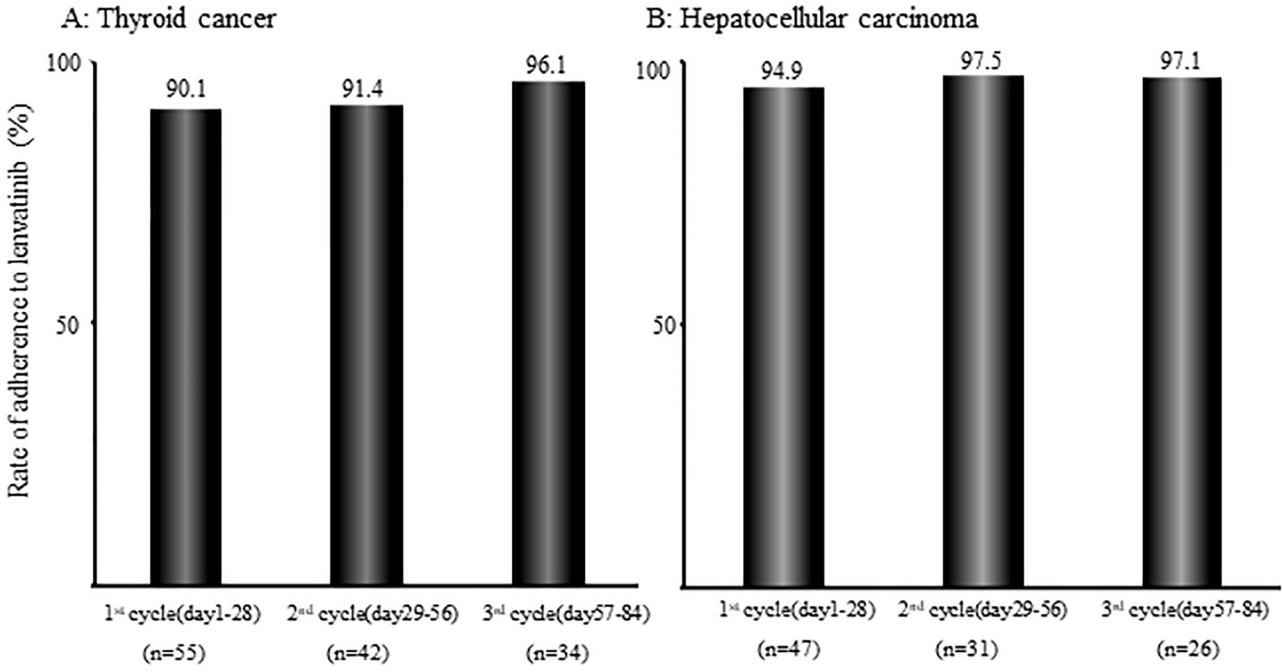

**Fig 2. Lenvatinib adherence rate in patients with thyroid cancer (A) and hepatocellular carcinoma (B) during cycles 1 to 3.** Adherence to lenvatinib was assessed using pill counting and self-reporting at the pharmaceutical outpatient clinic.

hepatocellular carcinoma] and diarrhea [3.9% (10 events) for thyroid cancer and 17.5% (17 events) for hepatocellular carcinoma].

The factors identified as contributing to lenvatinib non-adherence are shown as a bubble chart in Fig 4, with the number of non-adherence events on the vertical axis and the number of patients on the horizontal axis. In thyroid cancer, bleeding-related events showed a higher frequency and were associated with a larger number of non-adherent patients, as compared to other factors. On the other hand, the number of patients with decreased appetite was small, although the number of non-adherence incidents was high. In hepatocellular carcinoma, hypertension was associated with a greater number of non-adherent patients compared to other factors. Although the numbers of patients were small, diarrhea, coldness, fatigue, and nausea, in that order, were the next most frequent causes of decreased adherence.

## Discussion

In this study, adherence was assessed using a combination of two methods: pill counting and self-reporting. The results showed that adherence rates to lenvatinib for patients with refractory thyroid cancer and patients with unresectable hepatocellular carcinoma both exceeded 90% in this real-world context. The relative dose intensity was 66.3% for refractory thyroid cancer and 76.2% for unresectable hepatocellular carcinoma. The most common reason for decreased adherence to lenvatinib was bleeding events in thyroid cancer patients. Other factors included hand-foot skin reaction, diarrhea, hypertension, and edema. This is the first report on factors contributing to lenvatinib non-adherence in patients with thyroid cancer and hepatocellular carcinoma.

We found that hypertension, diarrhea, and fatigue were the main factors contributing to lower adherence to lenvatinib. Fig 4 shows that hypertension makes a relatively small

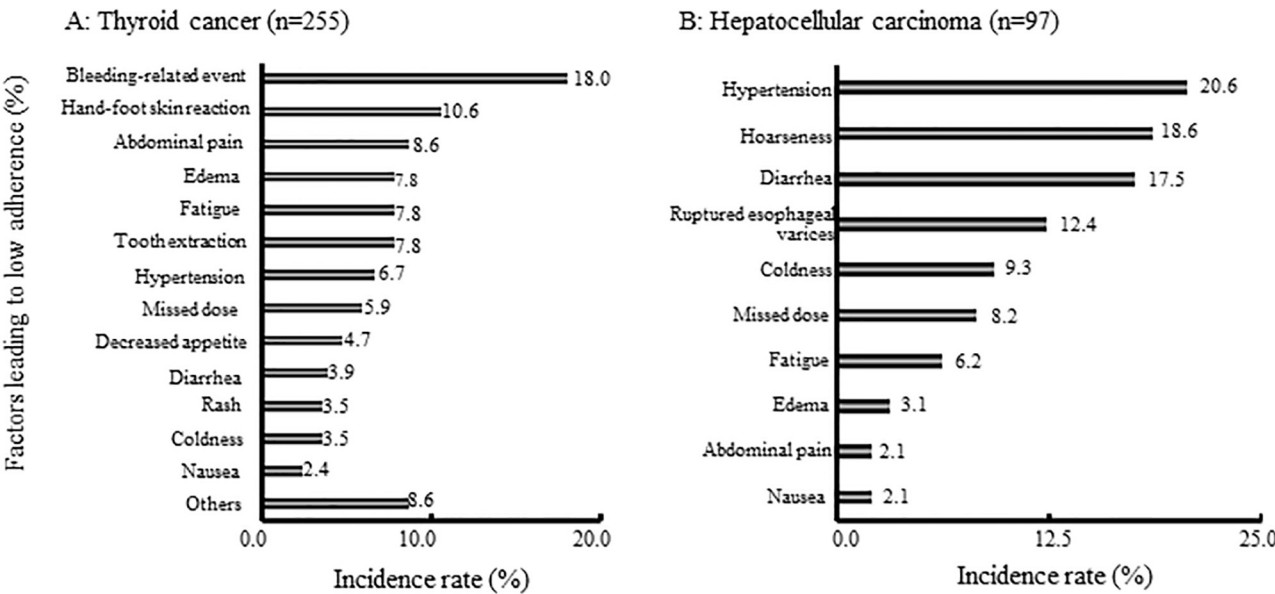

**Fig 3. Factors reducing adherence to lenvatinib in patients with thyroid cancer (A) and hepatocellular carcinoma (B).** In thyroid cancer, there were 255 incidents of lenvatinib non-adherence. In hepatocellular carcinoma, there were 97 incidents of lenvatinib non-adherence. The reasons for non-adherence were obtained by interview at the pharmaceutical outpatient clinic.

contribution to lower adherence per capita, but is a factor lowering adherence in many patients; indeed, more than 20% of patients presented with hypertension within 2 weeks after starting lenvatinib. In a subgroup analysis, more than 60% of Japanese patients with lenvatinib-induced hypertension presented with hypertension within 2 weeks of starting treatment

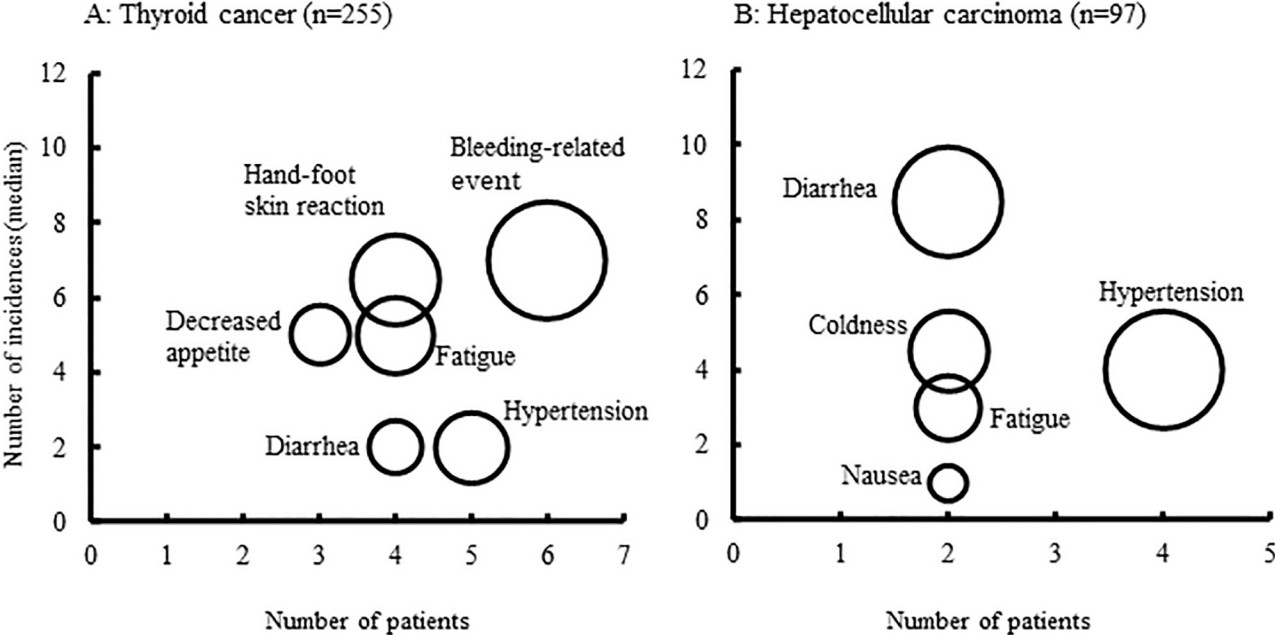

**Fig 4. Bubble chart of factors reducing adherence to lenvatinib in patients with thyroid cancer (A) and hepatocellular carcinoma (B).**

[16]. It seems likely that some patients experience a sudden rise in blood pressure at home, leading them to stop lenvatinib. Appropriate measures for the early control of hypertension may permit higher-dose treatment with lenvatinib, maximizing its therapeutic effect [17]. In most cases, hypertension can be controlled with antihypertensive drugs. Thus, prescribing antihypertensive drugs in advance and explaining their use to patients in advance may help avoid poor adherence to lenvatinib.

Diarrhea was given as a reason for non-adherence in 3.9% of patients with thyroid cancer and in 17.5% of patients with hepatocellular carcinoma. Diarrhea has been reported to be associated with survival in differentiated thyroid cancer [18]. Thus, antidiarrheal agents such as loperamide should be used as appropriate. Bleeding was also found to contribute to lower lenvatinib adherence in patients with thyroid cancer, but not hepatocellular carcinoma. The reason for this may be that lenvatinib is often used in thyroid cancer patients with postoperative recurrence. The main reported indications for dose reduction or withdrawal of lenvatinib are urinary protein, fatigue, and thrombocytopenia [8], but from the perspective of factors that decrease adherence, bleeding appears to be significant in thyroid cancer patients.

There is no gold-standard measurement method for oral medication adherence [5, 19], and it has been suggested that medication adherence should be measured in two ways [20, 21]. In this study, pill counting, an objective evaluation method, and self-reporting, a subjective evaluation method, were combined. Medication event-monitoring systems have been reported to be highly objective in assessing oral drug adherence [22]. However, they are not suitable for oral anticancer drugs, which require close monitoring of side effects and regular patient visits. Therefore, a combination of the two methods, pill counting and self-reporting, was considered appropriate for the present study. To improve adherence, pharmacists should assume a primary role in counseling, monitoring, and measuring adherence in clinical care [23]. First, it would be important for pharmacists to assess adherence to oral anticancer drugs in the clinical setting, using appropriate assessment methods. Then, pharmaceutical care and advice should be given to improve patient adherence.

The RDI for lenvatinib in thyroid cancer in the prior study was 71.2%, and the RDI for hepatocellular carcinoma was 87.7% in the 8 mg group and 87.5% in the 12 mg group [8, 9]. Although the RDI of lenvatinib in this study is lower than in the previous studies, it is not considered to be particularly low, considering that clinical trials enroll patients with good general health, renal function, liver function, and absence of other conditions. Although the RDI in this study was lower than in previous studies, this was due to the difference between clinical trials and actual clinical practice, and it cannot be considered that the RDI is extremely low. The present study examined not only the median relative dose intensity, but also the median adherence rate determined by pharmacists. The RDI reflects the physician's decision to withdraw or reduce the dose of lenvatinib, but adherence is important for oral anticancer drugs to ensure that patients take their medication at home. In addition, there was no difference in adherence rates between treatment lines.

In this study, lenvatinib adherence rates for thyroid and hepatocellular carcinoma exceeded 90% from one to three courses, suggesting that adherence is higher than with other molecularly targeted agents such as regorafenib. The median regorafenib adherence rate was reported to be 61.7% in the first cycle [24]. It has been reported that maintaining RDI in the first two months of lenvatinib treatment correlates with treatment efficacy [25], and patient care should be designed to ensure that adherence is well maintained in the first two months of treatment.

This study has several limitations. Firstly, it was conducted at a single institution and included only about 60 patients with each cancer type. However, it is important to note that this study focused on adherence to lenvatinib, a molecularly targeted drug, for two types of cancer. Furthermore, the study was conducted by a pharmacist who is a drug specialist at a

pharmaceutical outpatient clinic. Secondly, patients in this study were all Japanese. In the case of osimertinib, the incidence of interstitial lung disease has been reported to be higher in Japanese patients than in Western patients [26, 27]. Furthermore, regorafenib is also known to cause more severe side effects in Japanese patients than in Western patients [28], and this is also the case for lenvatinib. Thus, our findings may not be applicable to non-Japanese patients. Thirdly, the extent to which the decrease in adherence to lenvatinib and the decrease in RDI affect treatment outcomes was not investigated, and this is an issue for future studies.

In conclusion, the adherence rate for lenvatinib in thyroid and hepatocellular carcinoma in real-world clinical practice was more than 90% in Japanese patients in this study. Reasons for non-adherence were side effects of lenvatinib, such as hypertension, hand-foot skin reactions, and diarrhea. Notably, hypertension contributed to decreased adherence in many patients. Thus, patients should be informed in advance that their blood pressure may rise suddenly after starting lenvatinib and that antihypertensive medication is available if required. Such precautions should increase lenvatinib adherence. We believe the information obtained in this study will be useful in clinical practice to improve patient care.

## Author Contributions

**Investigation:** Minori Teramae.

**Supervision:** Kazuyoshi Kawakami, Naoki Fukuda, Takashi Yokokawa, Kazuo Kobayashi, Naoki Shibata, Wataru Suzuki, Hisanori Shimizu, Shunji Takahashi, Masato Ozaka, Naoki Sasahira, Satoko Hori, Masakazu Yamaguchi.

**Writing – original draft:** Yoshikazu Tateai, Kazuyoshi Kawakami.

**Writing – review & editing:** Yoshikazu Tateai, Kazuyoshi Kawakami, Satoko Hori.

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
