## [Decision Letter · Decision Letter 0]

17 Jul 2023

PONE-D-23-00635Factors associated with lenvatinib adherence in thyroid cancer and hepatocellular carcinomaPLOS ONE

Dear Dr. tateai,

Thank you for submitting your manuscript to PLOS ONE. After careful consideration, we feel that it has merit but does not fully meet PLOS ONE’s publication criteria as it currently stands. Therefore, we invite you to submit a revised version of the manuscript that addresses the points raised during the review process.

We look forward to receiving your revised manuscript.

Kind regards,

Shigao Huang

Academic Editor

PLOS ONE

Journal Requirements:

Reviewers' comments:

Reviewer's Responses to Questions

**Comments to the Author**

1. Is the manuscript technically sound, and do the data support the conclusions?

Reviewer #1: Partly

Reviewer #2: Yes

2. Has the statistical analysis been performed appropriately and rigorously? 

Reviewer #1: No

Reviewer #2: Yes

3. Have the authors made all data underlying the findings in their manuscript fully available?

Reviewer #1: Yes

Reviewer #2: Yes

4. Is the manuscript presented in an intelligible fashion and written in standard English?

Reviewer #1: Yes

Reviewer #2: Yes

5. Review Comments to the Author

Reviewer #1: Y. Tateai et al. have reported about factors associated with lenvatinib adherence in thyroid cancer and hepatocellular carcinoma. However, there are several concerns in this study to be published as a research article.

Major

#1 Lack of stratification and adjusted analysis

Authors performed adherence analysis together with different dosages for different carcinomas. If this is an accurate adherence study, it is necessary to add the data stratified with carcinoma types and treatment line, and to show criteria for dose reduction and withdrawal. If these factors related to the dose setting, the analysis adjusted for confounding factors is necessary.

#2 Definition of adherence rate and discussion

The definitions of adherence rate and RDI need to be clarified. In particular, the definition of adherence rate was unclear.

For example, if RDI with 8mg /5days per a week (weekend rest dosage) is different from 4mg/a day. How would authors indicate adherence rate in these cases?

In conclusion, authors described that hypertension was a major reason for non-adherence, followed by hand-foot skin reactions and diarrhea.

While adherence is usually related to subjective, the prescribed dose is influenced by not only subjective but also objective such as proteinuria, liver injury, and so on. Adherence may be easier to maintain if the dose is decreasing. How do authors discuss about this point?

#3 Carcinoma types

In clinical practice, lenvatinib has been approved for radioiodine-refractory thyroid cancer, unresectable hepatocellular carcinoma, advanced endometrial cancer, advanced renal cell carcinomas, and advanced or metastatic thymic carcinoma.

Why did authors focus on thyroid cancer and hepatocellular carcinoma? The reason why it is not a single cancer should also be stated.

#4 Unclear strength of this study

It was difficult to understand the strength and clinical application of this manuscript, and it is necessary to show originality in this study.

It would be a better paper if it could be discussed scientifically based on results of this study rather than speculation.

For example, authors described that “In this study, pill counting, an objective evaluation method, and self-reporting, a subjective evaluation method, were combined” in discussion. Authors may consider comparing these methods to show their usefulness.

Otherwise, authors described that “The RDI reflects the physician's decision to withdraw or reduce the dose of lenvatinib, but adherence is important for oral anticancer drugs to ensure that patients take their medication at home”. Authors may investigate how outcomes changes with RDI or adherence evaluations.

Minor

#1

Figure legend and table should be shown separated from the main text for easier reading.

It is difficult to distinguish where the main text continued if figure legend and table are included in the main text.

Reviewer #2: In the present paper Dr Tateai and coll. evaluate the lenvatinib adherence in 102 patients affected by thyroid or hepatocellular carcinoma in order to identify some factors associated with decreased lenvatinib adherence

I would like to congratulate the authors because the paper is clear and well written and the study is well conducted. The topic is interesting

Some suggestions:

1. Page 7 lines 128-12o About the therapeutic lines. Do you refer to both cases ( thyroid and hepatocellular carcinoma') Could you split these information ? Maybe you could add how many patients in first in third and further for hepatocellular carcinoma and how many in thyroid cancer in the table. Lenva for Hepatocellular carcinoma should be in first line.

2. If we look to the therapeutic line, do you find any difference in the adherence according this point?

3. About hypertension, do you provide any "prophylaxis" for acute adverse events?

4. I would like to ask if you have also outcome results in order to understand if there are some differences in the outcome according to therapeutic adherence! I think that this could add a great value to the paper

6. PLOS authors have the option to publish the peer review history of their article (what does this mean?). If published, this will include your full peer review and any attached files.

Reviewer #1: No

Reviewer #2: No

---

## [Author Response · Author response to Decision Letter 0]

16 Oct 2023

We would like to thank you for your advice on our manuscript. We have revised the manuscript according to the Reviewers’ comments. Please note that we have responded to all reasonable suggestions from the Reviewers.

---

## [Editor Report · Decision Letter 1]

31 Oct 2023

Factors associated with lenvatinib adherence in thyroid cancer and hepatocellular carcinoma

PONE-D-23-00635R1

Dear Dr. tateai,

We’re pleased to inform you that your manuscript has been judged scientifically suitable for publication and will be formally accepted for publication once it meets all outstanding technical requirements.

Kind regards,

Shigao Huang

Academic Editor

PLOS ONE
---

## [Editor Report · Acceptance letter]

6 Nov 2023

PONE-D-23-00635R1 

Factors associated with lenvatinib adherence in thyroid cancer and hepatocellular carcinoma 

Dear Dr. Tateai:

I'm pleased to inform you that your manuscript has been deemed suitable for publication in PLOS ONE. Congratulations! Your manuscript is now with our production department. 

Kind regards, 

on behalf of

Dr. Shigao Huang 

Academic Editor

PLOS ONE